# Nicotianamine: A Key Player in Metal Homeostasis and Hyperaccumulation in Plants

**DOI:** 10.3390/ijms241310822

**Published:** 2023-06-28

**Authors:** Ilya V. Seregin, Anna D. Kozhevnikova

**Affiliations:** K.A. Timiryazev Institute of Plant Physiology, Russian Academy of Sciences, Botanicheskaya St., 35, 127276 Moscow, Russia; kozhevnikova.anna@gmail.com

**Keywords:** copper, iron, manganese, metal accumulation in plants, metal detoxification, metal homeostasis, metal remobilization, metal transport, nickel, nicotianamine, stress, zinc

## Abstract

Nicotianamine (NA) is a low-molecular-weight N-containing metal-binding ligand, whose accumulation in plant organs changes under metal deficiency or excess. Although NA biosynthesis can be induced in vivo by various metals, this non-proteinogenic amino acid is mainly involved in the detoxification and transport of iron, zinc, nickel, copper and manganese. This review summarizes the current knowledge on NA biosynthesis and its regulation, considers the mechanisms of NA secretion by plant roots, as well as the mechanisms of intracellular transport of NA and its complexes with metals, and its role in radial and long-distance metal transport. Its role in metal tolerance is also discussed. The NA contents in excluders, storing metals primarily in roots, and in hyperaccumulators, accumulating metals mainly in shoots, are compared. The available data suggest that NA plays an important role in maintaining metal homeostasis and hyperaccumulation mechanisms. The study of metal-binding compounds is of interdisciplinary significance, not only regarding their effects on metal toxicity in plants, but also in connection with the development of biofortification approaches to increase the metal contents, primarily of iron and zinc, in agricultural plants, since the deficiency of these elements in food crops seriously affects human health.

## 1. Introduction

An excess or deficiency of mineral elements can have an adverse effect on living organisms, including humans, which is currently a growing threat. Many plant species are capable of accumulating metals, and metal concentrations in plant organs can considerably exceed their concentrations in the environment, as a result of which plants are an important metal source in food chains.

The ability to accumulate metals can vary significantly between different plant species and even between different populations of the same species. Most plant species are excluders, which accumulate metals mainly in their root systems and are often characterized by relatively low tolerance [1]. However, there are biogeochemical provinces in which metallophyte floras, usually characterized by a high percentage of endemism, formed on metalliferous soils during evolution [2,3]. The species that make up such floras are often characterized by high metal tolerance and the ability to accumulate metals in their shoots, the metal concentration therein often exceeding that in the environment. About 760 metal hyperaccumulating species from 82 families are known to date [4,5,6,7]. Some of these species are considered promising candidates for use in phytoremediation [8] and phytomining technologies [9], which are aimed at cleaning up and/or recultivating polluted soils, or metal extraction / recovery from plant shoots for further use in metal-containing chemicals.

Among the metals, elements such as iron (Fe), zinc (Zn), nickel (Ni), copper (Cu) and manganese (Mn) are essential for plants, but at supraoptimal levels can have a toxic effect on various physiological processes, as reviewed in [10,11,12,13,14,15], which is manifested as the disruption of plant growth and morphogenesis, as reviewed in [16]. Therefore, it has recently been proposed to call them potentially toxic elements [17]. Under natural growth conditions, hyperaccumulators are capable of accumulating more than 300 μg/g Cu, 1000 μg/g Ni, 3000 μg/g Zn and 10,000 μg/g Mn in shoots per unit of dry mass, without visible disturbance of their metabolism and growth, which is much more than in non-hyperaccumulators [6,7]. Model plants for studying the phenomenon of hyperaccumulation are *Arabidopsis halleri* and *Noccaea caerulescens*. Significant intraspecific differences in metal tolerance and hyperaccumulation capacities have been identified in these species, which were associated with different edaphic and/or phylogeographic origins [18,19,20,21,22,23,24].

The mineral composition of plants, also called the ionome [25], is the result of the interaction between the endogenous processes in plants and the environment [26]. Plants use a variety of mechanisms for metal sequestration, exclusion, and chelation to maintain metal homeostasis and prevent toxic effects [14].

Despite the fact that the ability to hyperaccumulate evolved independently in *A. halleri* and *N. caerulescens*, both species acquired similar features: an increased metal uptake rate, efficient symplastic transport and limited accumulation of metals in root cell vacuoles, enhanced metal (un)loading, and an increased rate of metal transport to the shoot, where they are efficiently detoxified. These traits are generally characteristic of other hyperaccumulator species, as reviewed in [4,7,27,28,29,30]. An important role in metal transport, detoxification, and maintenance of metal homeostasis is played by low-molecular-weight ligands, the main function of which is to sustain the labile pool of metals [31,32]. If ligands form multidentate complexes with metals, they are called metal chelators [31]. One of such chelators is nicotianamine.

The study of the non-proteinogenic amino acid nicotianamine (NA, 2S:3S′:3S″-*N*-[*N*-(3-amino-3-carboxypropyl)-3-amino-3-carboxypropyl]-azetidine-2-carboxylic acid) was historically closely associated with its role in maintaining Fe homeostasis, as reviewed in [30,31,33,34,35,36,37]. Nicotianamine was first isolated from the leaves of *Nicotiana tabacum* (hence its name) in 1991 [38]. A few years later it was isolated from the seeds of *Fagus silvatica*, and its structure was first characterized [39]. It is composed of an azetidine ring followed by two aminobutyrate moieties (Figure 1) [40]. The role of NA as a “normalizing factor” involved in Fe homeostasis was first described in the study on the *chloronerva* (*chln*) mutant of *Solanum lycopersicum* (previously *Lycopersicon esculentum*), which was characterized by retarded growth, interveinal chlorosis of young leaves and infertility. It was found that a mixture of whole plant extract, even obtained from other species, or grafting with a wild-type plant was able to normalize this phenotype [41,42,43]. After this discovery, the interest in this small molecule greatly increased and, within a short time, NA was found in the representatives of various systematic groups of living organisms: vascular plants, mosses, club mosses, horsetails, and fungi [44]. Nicotianamine biosynthesis was observed in all studied plant species. It was shown that NA is involved in, next to Fe homeostasis, the maintenance of the homeostasis of Zn, Ni, Cu, and Mn, by binding them inside cells (Figure 1) [35,40,45,46,47,48,49,50,51]. In 2022, a previously unreported compound, dihydroxy-nicotianamine, was identified as the most abundant Cu and Zn ligand in hypertolerant *Hypericum laricifolium* from a post-mining area of the Peruvian Andes [51]. The data obtained in this work for the first time convincingly demonstrated the presence of hydroxyl (mono- and di-) derivatives of NA (OH-NA and di-OH-NA) in some plant species.

Due to the fact that Fe is usually present in the environment in an oxidized, trivalent, water-insoluble form and is poorly available, plants have developed two Fe absorption strategies: reduction- and chelation-based strategies, also known as Strategy I and Strategy II, respectively [11,52,53]. Non-graminaceous species utilizing Strategy I for Fe uptake secrete organic acids and protons to lower the pH of the rhizosphere, which increases the solubility of Fe (III), then convert ferric (Fe^3+^) iron to ferrous (Fe^2+^) iron using ferric chelate reductase (FCR, EC 1.16.1.7) and absorb Fe^2+^ through the roots, for example, via non-selective iron-regulated transporter 1 (IRT1), belonging to the ZIP (ZRT/IRT-like proteins) family [30,37,54,55]. Graminaceous species employ an enhanced method of Fe uptake known as Strategy II, using phytosiderophores (phytometallophores) which can effectively bind and increase the bioavailability of Fe, Zn, Cu, Mn, Ni, cadmium (Cd), and cobalt (Co) [30]. Phytosiderophores include 2′-deoxymugineic acid (DMA) and eight different compounds from the mugineic acid (MA) family, which are secreted by the root cells of cereals via the transporter of MAs (TOM1), belonging to the major facilitator superfamily (MFS) [30,53,55,56,57,58].

Previously, it was believed that phytosiderophores are characteristic only of cereals. However, DMA has recently been found in the xylem sap and leaves of *Olea europaea* [59], and in the roots and shoots of *Nicotiana thyrsiflora*, *Arenaria digyna*, and *Puya* sp. [51], while MA derivatives have been detected in root exudates of some dicot species [60,61], which indicates a wider distribution of these compounds. In addition, *A. halleri* is capable of secreting NA as a phytosiderophore, which resembles Strategy II mechanisms, although this species belongs to the Brassicaceae family of dicotyledons [62]. It is generally accepted that some species can utilize a hybrid strategy depending on soil pH and oxygen status [52]. To sum this up, more and more results indicate that the distinction between Strategy I and Strategy II plants (the latter as characteristic of graminaceous species only) used so far is most probably not so clear-cut.

The discovery of NA-like metallophores in bacteria testifies to the ancient origin of the mechanisms involved in the tolerance to high concentrations of metals that prevailed in the early history of the Earth [40]. Numerous studies on the biosynthesis of NA, the transport of NA and its complexes with metals, as well as the physiological functions of this small organic molecule are analyzed in this review. In addition, we discuss the involvement of NA in the mechanisms of metal hyperaccumulation, which is of fundamental and practical importance. We apologize in advance to all authors whose papers were not cited due to the limited scope of the review.

## 2. Formation of Metal–Nicotianamine Complexes

Nicotianamine can be found in all plant organs, with the highest concentration observed in the root and shoot meristematic tissues [35]. Nicotianamine forms a hexadentate complex with metal ions [33,46]. The stability of NA complexes with metals depends on the physicochemical properties of metal ions. Based on the stability constants (log *K*) of NA complexes with metal ions, their strength decreases in the following order: Fe^3+^ (20.6) > Cu^2+^ (18.6) > Ni^2+^ (16.1) > Co^2+^ (14.8) ≈ Zn^2+^ (14.7) > Fe^2+^ (12.1) > Mn^2+^ (8.8) [63,64].

Since NA plays a key role in the homeostasis of Fe, an element with variable valency, considerable attention has been paid to studying the formation of Fe–NA complexes [65,66,67]. A detailed study of the Fe–NA complexes via the methods of coordination chemistry showed that Fe(III)–NA is thermodynamically more stable in comparison with Fe(II)–NA, while Fe(II)–NA is kinetically more stable than Fe(III)–NA. Under the physiological conditions prevailing in plant tissues, Fe(III)–NA can undergo reduction, but the auto-oxidation of Fe(II)–NA to Fe(III)–NA is prevented. As a result, two types of Fe–NA complexes can be found in plant species [67]. However, it is not completely clear which of them predominates in the cytoplasm. Capillary electrophoresis showed that at a neutral pH, NA preferentially forms complexes with Fe(II) [65]. However, later it was found that NA has comparable affinities for Fe(II) and Fe(III) at a cytosolic pH (7.2–7.5) [68]. The kinetics of the complex formation and the stability of Fe(II)–NA and Fe(III)–NA complexes were also studied at a neutral pH using spectrophotometric, potentiometric and Fenton activity measurements, which showed that Fe–NA complexes were relatively poor Fenton reagents. Surprisingly, auto-oxidation was not observed when Fe^2+^ was complexed with NA, even during the deliberate oxygenation of the solution [65]. Therefore, the formation of stable metal complexes with NA may be one of the mechanisms that protect the cell from metal-induced oxidative stress.

Nicotianamine complexes with metals were identified in different plant species using various methods. For example, Zn–NA complexes were identified in the phloem sap of *Oryza sativa* [69] and *Ricinus communis* [70], in the xylem sap and embryo sac liquid of *Pisum sativum* [71], in the roots, stems, and leaves of *N. thyrsiflora* and *A. digyna* [51], in the leaves of *Lactuca sativa* and *Puya* sp. [51,72], and in *Cocos nucifera* [73]. The Cu(II)–NA complex was identified in the xylem sap of *P. sativum* [71] and in the phloem and xylem sap of *R. communis* [70], and was characterized in detail, for example, in *N. caerulescens* [74] and *C. nucifera* [73]. The ratio between the complexes of NA and other ligands formed with different metals can vary significantly [75]. Thus, the distribution of NA, OH-NA, di-OH-NA, and DMA complexes with Cu and Zn differed significantly between organs, metals, as well as plant species [51].

The stability of NA complexes with metal ions depends on the pH, which can play a decisive role in their formation in the cytosol (pH 7.2–7.5) as well as the vacuolar (pH 4.5–6.0), xylem (pH 5.0–6.2), or phloem (pH 7.0–8.0) sap [30,55,76]. The NA–Mn complexes started to form at pH > 6.2, and the Fe(II)-NA and Zn–NA complexes were completely formed at pH 6.2, while the Ni-NA and Cu(II)-NA complexes were formed at pH 5.2. The latter were even found at pH < 4.0. For all studied metals, the stability of the complexes with NA was maximal at pH 6.5 [35,76]. It is assumed that NA hydroxylation and the formation of OH-NA and di-OH-NA in *H. laricifolium* can increase metal–chelate stability in acid environments [51].

## 3. Biosynthesis and Accumulation of Nicotianamine in Plants

### 3.1. Biosynthesis of Nicotianamine

The precursor for NA biosynthesis is L-methionine, which is formed through the methionine cycle. Subsequently, S-adenosylmethionine is synthesized from L-methionine and ATP by S-adenosylmethionine synthetase (SAMS, EC 2.5.1.6) [53,77] (Figure 1). The prediction of protein subcellular localization showed that most of the SAMS proteins were localized in the cytoplasm or in the chloroplasts [78]. The biosynthesis of NA occurs, apparently, in a single-step reaction via the condensation of three S-adenosylmethionine molecules [30,40,79,80,81,82] by nicotianamine synthase (NAS, EC 2.5.1.43) (Figure 1), which was first isolated from *Hordeum vulgare* [80,81,83]. Cultivation of *H. vulgare* under Fe deficiency resulted in a 5-fold increase in NAS activity in roots [84], which provides evidence of a pivotal role of NA in the maintenance of Fe homeostasis.

### 3.2. The Expression of NAS Genes in Plant Organs

Genes encoding NAS have been found in the representatives of different kingdoms: archaea, bacteria, fungi (for example, *Neurospora crassa*), and plants [36,40]. Typically, 1–9 *NAS* genes are found in plant genomes, although 20–21 NAS genes were identified in the genome of hexaploid varieties of *Triticum aestivum* [85,86] (Table 1). 

The data obtained made it possible to carry out a phylogenetic analysis of the *NAS* genes from these species [85,93,100,105], which resulted in their division into two groups, clade I and clade II [85,100]. The gene encoding NAS was also identified in *Physcomitrella patens*, the only moss species whose genome has been deciphered [40].

The expression level of *NAS* genes can vary in different plant organs. In *Arabidopsis thaliana*, *AtNAS1* and *AtNAS4* are expressed in both roots and shoots, *AtNAS2* is predominantly expressed in roots, and *AtNAS3* is mainly expressed in leaves [89]. Two of the eight *TmNAS* genes from *Triticum monococcum* (clade II) were highly expressed in shoot tissues, while the other six (clade I) were expressed in roots [100]. In Fe-sufficient *O. sativa* plants, *OsNAS1* and *OsNAS2* expression was observed in roots, but not in leaves, while the *OsNAS3* transcript was present in leaves but was very low in roots [95,97]. In contrast to *O. sativa* and *Zea mays*, in *H. vulgare* all currently known *HvNAS* genes show Fe deficiency-inducible expression specifically in root tissues, with no expression detected either under Fe sufficiency or, in shoot tissues, under any Fe status [95]. The expression of *MxNAS1/2/3* was observed in the roots and leaves of *Malus xiaojinensis* [104,105,106], and the expression of, for example, *MxNAS3* was higher in roots and young leaves compared to that in old leaves under normal Fe conditions [105]. Although the possibility of arsenic (As) binding to NA in vivo was not shown, the expression of *RcNAS1* and *RcNAS2* increased in roots and leaves of the As(V)-treated (202 μM) As-tolerant genotype and decreased in the As-sensitive genotype of *R. communis*. *RcNAS3* expression was decreased in leaves and increased in roots in both genotypes under As treatment [94].

Different metal and nitrogen concentrations in the growth medium can affect the expression level of *NAS* genes. Increased levels of *NAS* genes expression were observed under Fe deficiency in the roots of *H. vulgare* [81] and *T. aestivum* [85], in the roots of Cd- and Zn-treated *Sedum alfredii* [92], in the roots of Cd-treated *Triticum turgidum* [108], in Cd-treated *N. caerulescens* [109], in the leaves of *Arachis hypogaea* under excess Mn [110], and in *A. thaliana* under a deficiency of Cu (*NAS1* in leaves), Zn (*NAS1/2/3* in roots) and Fe (*NAS1/3* in roots) [111], as well as after foliar Zn application in *Medicago sativa* (*NAS1* in shoots) [112]. In *T. aestivum*, high nitrogen levels in the medium led to increased expression levels of *TaNAS1* and *TaNAS2*, which were accompanied by increased Fe and Zn root-to-shoot translocation [113].

The expression of *NAS* genes in one species can be differently regulated under metal deficiency and excess, which indicates a different role of NA under diverse conditions. For example, under Fe deficiency, a significant increase in *OsNAS1* and *OsNAS2* expression but not in *OsNAS3* expression was observed, whereas under Fe excess, *OsNAS1* and *OsNAS2* expression was suppressed in roots, and *OsNAS3* expression was significantly increased in roots, stems, young and, particularly strongly, in old leaves of *O. sativa* [98,99]. In contrast, under Zn deficiency, *OsNAS3* expression was significantly increased in almost all tissues [99]. Under Zn excess, *OsNAS3* expression was highly repressed in both roots and shoots, while *OsNAS1* and *OsNAS2* were highly induced in roots [114]. *ZmNAS1* and *ZmNAS2* were expressed only in Fe-deficient roots of *Z. mays*, while *ZmNAS3* was expressed under Fe-sufficient conditions and was negatively regulated by Fe and Zn deficiency [101,115]. In *T. monococcum,* root-specific *TmNAS* genes were up-regulated under Fe or Zn deficiency, whereas shoot-specific *TmNAS* genes were up-regulated under Fe or Zn sufficiency [100].

### 3.3. The Expression of NAS Genes in Plant Tissues

In different root and shoot tissues, the expression level of *NAS* genes may vary depending on Fe levels, which was studied most extensively in *O. sativa* and *Z. mays*. Promoter–GUS analysis revealed that *OsNAS1* and *OsNAS2* were expressed in the companion cells and pericycle cells adjacent to the protoxylem in Fe-sufficient roots, whereas under Fe deficiency, their expression extended to all root tissues and was found in the vascular bundles of green leaves, which may be related to the involvement of NA in Fe uptake and redistribution [97]. Under Fe excess, *OsNAS3* expression was particularly prominent in the exodermis, rhizodermis and vascular bundles, and extremely strong expression was observed in the phloem cells, phloem companion cells, protoxylem, xylem parenchyma cells, and rhizodermal cells of *O. sativa* roots [99]. In contrast, under Fe deficiency, *OsNAS3* expression was limited to the tissues of the root central cylinder [97]. In shoots under Fe excess, the expression of this gene was observed mainly in the vascular bundles as well as in leaf trichomes, which may be associated with metal detoxification in the epidermal cells [99]. In addition, regardless of the Fe status, a strong expression of *OsNAS3* was localized to the discrimination center that includes the shoot meristem, node, and internode [98,99]. All of these genes were expressed in germinating grains of *O. sativa*, indicating the involvement of NA and DMA in metal transport during germination [116]. Using in situ hybridization, it was revealed that *ZmNAS1;1/1;2* genes were mainly expressed in the root cortex and stele under Fe sufficiency, and their expression was expanded to the rhizodermis, as well as to the shoot apices under Fe deficient conditions. On the contrary, *ZmNAS3* was expressed in the axillary meristems, leaf primordia and mesophyll cells [103]. In dicotyledonous plants, *NAS* gene expression was also often detected in conducting tissues. For example, the expression of *MxNAS1/2*/*3* was observed to a greater extent in the phloem than in the xylem of *M. xiaojinensis* [104,105,106], and the expression of *MtNAS2* was found in the xylem parenchyma cells of *Medicago truncatula* [117], which may be related to the involvement of NA in long-distance metal transport.

### 3.4. Regulation of NA Biosynthesis

To adapt to a changeable environment, plants have a set of sophisticated regulatory systems at the transcriptional and post-transcriptional levels. The expression of *NAS* genes is regulated by various transcription factors. In *A. thaliana*, the basic helix-loop-helix (bHLH) fer-like iron deficiency-induced transcription factor (FIT) induces the expression of transcription factor genes *MYB10* and *MYB72*, which positively regulate the expression of the *AtNAS4* gene [118] as well as other genes induced by Fe deficiency [37]. In the *myb10myb72* double mutant, no *AtNAS4* gene transcript accumulation was observed, which was accompanied by a reduced Fe content in plants, as well as Ni and Zn sensitivity similar to that of the *nas4* mutant [118]. Another bHLH-type transcription factor, POPEYE (PYE), can also regulate the expression of Fe deficiency-responsive genes. Under Fe deficiency, *PYE* expression is strongly induced in the pericycle. The loss of PYE function causes a significantly increased and prolonged expression of *AtNAS4* after exposure to Fe deficiency. ChIP-on-chip analysis indicates that *AtNAS4* promoters are directly bound by PYE, which directly represses the induction of *AtNAS4* under Fe deficiency [119]. In *O. sativa*, iron-regulated transcription factor 3 (OsIRO3), which is a homologue of PYE, can directly bind to the E-box in the promoter of *OsNAS3*, thus negatively regulating its expression in response to Fe deficiency [120]. It was also observed that the transcription factor TmbHLH47 directly interacted and promoted the transcription of *TmNAS3* in *T. monococcum* [121]. Thus, both FIT and PYE interact with various bHLH proteins, and these two networks are jointly involved in Fe homeostasis [120,122]. In *A. thaliana*, the F-group basic region leucine-zipper transcription factors bZIP19 and bZIP23 function simultaneously as the sensors of intracellular Zn status, via the direct binding of Zn ions to a Zn sensor motif, and as the central regulators of the Zn deficiency response, with their target genes including Zn transporters from the ZIP family and NAS enzymes [123,124]. A similar regulatory mechanism involving MtFbZIP1, which is a functional homologue of bZIP19 and bZIP23 from *A. thaliana*, was found in *M. truncatula* [125]. NAC transcription factors make up one of the largest families of transcriptional regulators in plants. Members of this gene family have been suggested to play an important role in the regulation of the transcriptional reprogramming involved in the plant stress response. Genome-wide analyses of loss- and gain-of-function mutants revealed that OsNAC6, whose gene is mainly expressed in the root endodermis, pericycle and phloem, up-regulated the expression of *OsNAS* genes and NA biosynthesis in *O. sativa* roots, thereby conferring plant tolerance to drought [126]. It is important to note that the transcription level of the *AhNAS2* gene was almost the same when *A. halleri* plants were grown in hydroponics and in soil [127], which suggests that the data obtained under laboratory conditions can be extrapolated to plants from natural populations.

Relatively little is known about the hormonal regulation of *NAS* gene expression. The expression of *MxNAS1/2/3* was highly affected by indole-3-acetic acid treatment, whereas only *MxNAS3* was greatly influenced by abscisic acid treatment in *M. xiaojinensis* seedlings [104,105,106]. The expression of *ZmNAS* genes in *Z. mays* seedlings was regulated by jasmonic acid, abscisic acid, and salicylic acid [102]. Fe deficiency enhanced ethylene production and signaling in *O. sativa*. RNA interference of *OsIRO2* in transgenic plants showed that ethylene acted via this transcription factor to induce the expression of *OsNAS1* and *OsNAS2*. By contrast, ethylene was not involved in the Fe deficiency response in *H. vulgare* [128]. 

### 3.5. Nicotianamine Accumulation in Metal Hyperaccumulators and Excluders

A change in *NAS* expression level results in a change in the content of NA in plant organs. In *O. sativa* plants with an increased expression of the *OsNAS3* or *OsNAS2* genes, an increased NA level was observed, which was accompanied by Zn accumulation in leaves and grains [129,130], while the increased expression of *OsNAS1* resulted in increased NA, DMA, and Fe levels in the endosperm and embryo, and elevated the Zn content only in the endosperm [131]. Constitutive expression of *OsNAS2* in *T. aestivum* up-regulated the biosynthesis of NA and DMA, which was also accompanied by Fe and Zn accumulation in the grain [132]. Under Fe deficiency, *MxNAS1* and *MxNAS3* expression was enhanced to stimulate NAS and NA biosynthesis [105,133], which facilitated Fe acquisition by *M. xiaojinensis* in an Fe-depleted environment [104]. The NA content in *M. xiaojinensis* mature leaves was higher than that in stems, young leaves, and roots under both Fe deficiency and Fe sufficiency conditions [133]. Conversely, *NAS2* gene silencing in *A. halleri* led to a decrease in the NA content and, as a result, to a significant decrease in the Zn content in leaves [127,134]. In *nas124* and *nas236* triple mutants of *A. thaliana*, lower quantities of NA were observed, which was followed by a lower Fe level in the nucleolus in *nas124* plants [135]. 

Hyperaccumulators and excluders can differ significantly in the levels of *NAS* gene expression. The expression levels of *AhNAS2/3/4* in *A. halleri* and *NcNAS2/4* in *N. caerulescens* were constitutively much higher compared to those of the closely related excluders *Arabidopsis lyrata*, *A. thaliana*, and *Thlaspi arvense*, resulting in an elevated NA content in the roots of the hyperaccumulators and indicating an essential role of NA in the mechanisms of hyperaccumulation [5,62,87,88,90,91,134,136,137]. In addition to the elevated levels of *NAS* gene expression, high levels of *SAMS1-3* gene expression in *A. halleri* root tissues may contribute to the high NA concentration therein [136].

The expression level of *NAS* genes and the NA content in hyperaccumulators vary with the metal concentrations in the environment. In the Cd/Zn hyperaccumulator *S. alfredii, SaNAS1* was highly expressed under both Cd and Zn treatments [93], while the *SaNAS2-5* expression levels were all upregulated under exposure to 50 μM Cd, and the expression of *SaNAS2* was significantly upregulated in roots under exposure to 500 μM Zn [92]. The metal-induced increase in *SaNAS1* expression was accompanied by an increase in the NA levels in the roots of *S. alfredii*, and the expression of *SaNAS1* in *A. thaliana* mutants increased NA production and promoted Cd and Zn accumulation in the roots and shoots of the mutants, as well as their metal tolerance [93]. The content of NA in *N. caerulescens* increased with the concentration of Ni [138] and Zn [139] in the medium. 

Different populations of *N. caerulescens* can vary in the level of *NAS* gene expression, which may partly determine their different abilities to accumulate metals [140,141]. In the absence of Ni, the expression levels of *NcNAS1* and *NcNAS3* were higher in the plants of La Calamine population from Zn-enriched calamine soil than in the plants of Monte Prinzera population from Ni-enriched ultramafic soil, whereas in the presence of Ni (10 μM) the expression levels of various *NAS* genes were significantly increased only in the latter [141]. Meanwhile, the Ni concentration in Monte Prinzera leaves was significantly higher than that in La Calamine [141], which was also observed at a lower Ni concentration in the medium (1 µM) [24].

The ratio of the NA levels in different plant tissues can vary considerably. In hyperaccumulators, metal accumulation in the pavement epidermal cells and trichomes can be considered one of the most important detoxification mechanisms [30,142]. However, using a proteomic approach, it was shown that in the epidermis of *N. caerulescens*, the proportion of Zn–NA complexes was small and Zn was mainly bound to citric and malic acids, while in the mesophyll a significant proportion of Zn was bound to NA [143], which is an important component of metal homeostasis maintenance and leads to a reduction in Zn toxic effects on photosynthesis under metal excess.

### 3.6. Localization of NAS Proteins in Metal Hyperaccumulators and Excluders

The intracellular localization of different NAS proteins can differ considerably, although the experimental data on this subject are debatable. A bioinformatics study of the ZmNAS protein sequences revealed signal peptides localizing ZmNAS5 (clade II) and ZmNAS6;1 (clade I) to mitochondria and chloroplasts, respectively [102]. However, in another work, the same authors conclude that in *Z. mays* all ZmNAS proteins are localized in the cytoplasm, suggesting that the variable N-terminal domain has little effect on subcellular localization and NA is synthesized in the cytoplasm, at least in *Z. mays* [103]. In Fe-deficient *O. sativa*, OsNAS2 enzyme activity was found to be localized in root vesicles [144], and MxNASs were mainly localized in the plasma membrane [104,105,133]. In the hyperaccumulator *S. alfredii*, SaNAS1 protein was distributed throughout the cytoplasm and nucleus [93], which, together with other data, may indicate the possibility of NA biosynthesis in various cellular compartments.

The localization of NAS proteins at the tissue level can also be heterogeneous, which may in part influence the accumulation of NA in the cells of different tissues. For example, immunohistochemical staining showed that MxNAS1 was localized mainly in the rhizodermal cells, in the root and stem conducting tissues, as well as in the palisade mesophyll of mature leaves and in the parenchyma cells of young leaves of *M. xiaojinensis*, with protein levels being higher in all tissues under Fe deficiency compared to those under Fe-sufficient conditions [133].

### 3.7. Nicotianamine as a Precursor of Phytosiderophores in Cereals

In cereals, utilizing the second Fe uptake strategy involving phytosiderophores, nicotianamine aminotransferase (NAAT, EC2.6.1.80) catalyzes the transfer of the NA amino group to form the corresponding 3″-keto intermediate, which is converted into DMA by 2′-deoxymugineic acid synthase (DMAS, EC:1.1.1.285). Phytosiderophores of the MA family are subsequently formed from DMA [30,55,82,145,146].

Thus, it is assumed that clade I NAS genes (e.g., *NAS1* and *NAS2*) function primarily in the regulation of NA and MAs biosynthesis, Fe uptake and long-distance Fe transport, are preferentially expressed in root and stem tissues and are up-regulated in response to Fe deficiency. Conversely, clade II NAS proteins (e.g., ZmNAS3 and OsNAS3/4/5) do not contribute to the biosynthesis of MAs under Fe deficiency and are instead primarily involved in NA biosynthesis for Fe loading into the conducting tissues and the maintenance of cellular Fe homeostasis. They are mainly expressed in the leaf and are up-regulated in response to Fe excess [85,100]. The expression of the genes from the two clades may be differentially regulated under fluctuating Fe levels [58]. The NA-to-DMA ratio plays a key role in maintaining the homeostasis of not only Fe, but also of Zn, Mn, and Cu in cereals [50,58,131].

## 4. Transport and Physiological Role of Nicotianamine in Plants 

### 4.1. The Role of Yellow Stripe-like Transporters in Metal Transport in Plants 

Recently, large amounts of the data on the molecular mechanisms of transport of NA and its complexes with metals throughout the plant have appeared. Yellow stripe-like (YSL) transporters, which belong to the oligopeptide transporter (OPT) superfamily, are involved in the uptake, transport and retranslocation of metal complexes with NA and phytosiderophores [147,148]. There are two groups of YSL transporters, some of which are involved in the uptake of metal complexes with phytosiderophores in cereals, while the others are involved in the redistribution and long-distance transport of metal complexes with NA in monocots and dicots [149] (Figure 1).

YSL transporters are located primarily in the plasma membrane, with an average of 12–14 domains traversing it [149,150,151,152,153,154,155,156,157,158], and have different substrate specificities (Table 2). 

Stable Fe(III) complexes with phytosiderophores are taken up by root cells via the YS1 transporter, which was first discovered in *Z. mays* in 2001 [47,169]. ZmYS1 from the roots and leaves of *Z. mays* has wide substrate specificity and is able to transport complexes of phytosiderophores with Fe(III), Fe(II), Cu(II), Mn(II), Ni, Zn, and Cd, as well as Fe(II)/Fe(III)/Ni–NA in symport with protons [170,171]. Recently identified ZmYSL2, which is highly homologous to ZmYS1, is involved in Fe–NA and Zn–NA transport [172,173,174]. However, the HvYS1 transporter from *H. vulgare*, which is also very similar to ZmYS1, is involved only in Fe(III)–DMA and Fe(III)–MA transport [160,161,162]. The substrate specificity of HvYS1 is determined by an extracellular loop located between transmembrane domains 6 and 7 [161]. Unlike HvYS1, HvYSL2 has broader substrate specificity and is able to transport Fe(III)/Zn/Mn/Cu/Co/Ni–DMA and Fe(II)–NA [154].

The distribution of YSL transporters at the organ and tissue levels can differ significantly (Table 2). Under Fe deficiency, the expression of the *ZmYS1* gene was observed not only in the root rhizodermal cells, but also in the shoots [35,169], whereas the expression of the *HvYS1* gene was detected only in the rhizodermis [160], and the expression of the *HvYSL2* gene was found in the root endodermis and shoots [154]. ZmYSL2 is specifically localized in the plasma membrane facing the maternal tissue of the basal endosperm transfer cell layer and functions in loading Zn–NA into these cells [172,174]. In addition, ZmYSL2 is preferentially accumulated in aleurone, sub-aleurone, and embryo cells, which is associated with its participation in Fe and Zn transport from the endosperm into the embryo during kernel development, as well as in aleurone cell fate specification and the proper functioning of starchy endosperm cells [172,173]. The expression level of *ZmYS1* increased both in roots and shoots under Fe deficiency and was not regulated by Cu or Zn deficiency [169,170]. The *ys1* mutant of *Z. mays* with a reduced level of *ZmYS1* expression showed all signs of Fe deficiency even at a sufficient level of Fe in the medium [57] and absorbed less Zn–DMA compared to the wild-type plants [184].

In *O. sativa*, the family of YSL transporters includes 18 representatives [35] involved in the transport of various metal–ligand complexes and these transporters are localized at various tissues and organs. Only a few of them have been described in detail (Table 2). For example, OsYSL2, whose gene is expressed under Fe deficiency in the phloem cells, especially in the phloem companion cells in leaves, and in the epithelium, the vascular bundle of the scutellum, and the leaf primordium in mature grains, transports Fe(II)–NA and Mn(II)–NA from roots to shoots and caryopses along the phloem [116,150,163]. OsYSL6, whose gene is expressed in roots and shoots regardless of the deficiency or toxicity of different metals, is involved in the transport of Mn(II)–NA [153]. OsYSL9, whose gene expression is induced in the root vascular cylinder but repressed in the non-juvenile leaves in response to Fe deficiency, as well as observed in the scutellum of the embryo and in the endosperm cells surrounding the embryo, is involved in Fe(III)–DMA and Fe(II)–NA transport from the endosperm to the embryo in developing caryopses [158,164]. OsYSL13, whose gene is highly expressed in leaves, especially in leaf blades, and is induced by Fe deficiency, both in the root cortex and in shoots, is involved in Fe distribution [151,156]. OsYSL15, whose gene expression is also induced by Fe deficiency in almost all root tissues, is mainly involved in Fe(III)–DMA and Fe(II)–NA uptake, and, at a sufficient level of Fe, is also involved in Fe transport along the phloem [151,158,165]. OsYSL16, whose gene is expressed in the rhizodermis and the root and shoot conducting tissues, as well as in the rachilla, palea, lemma, anther and ovary, transports Fe(III)–DMA and Cu–NA, participating in the long-distance transport and redistribution of these metals to the floral organs [166,167,168]. Initially, OsYSL18 was shown to transport only Fe(III)–DMA, but not Fe(II)–NA, Zn–DMA or Zn–NA, and its gene expression was shown both in generative organs, including the pollen tube, and in vegetative organs, in lamina joints, the inner cortex of crown roots, and the phloem parenchyma and companion cells at the basal part of every leaf sheath [152]. However, it was later suggested that Fe–NA could be transported into the shoots by OsYSL17 and OsYSL18, whose genes were induced by Fe excess in the roots and the discrimination center [98]. Other YSLs, such as *OsYSL13* and *OsYSL14*, were expressed in both leaves and roots, whereas *OsYSL5/7/8/12/17* were expressed in roots under both Fe-sufficient and Fe-deficient conditions [150,151]. Increased expression of *OsYSL15* resulted in an increased Fe content in the leaves and seeds of transgenic plants, indicating the involvement of OsYSL15 in Fe uptake and transport in *O. sativa* [165].

The highly conserved nature of YSLs provides an opportunity to extrapolate the functionality of YSL proteins from *O. sativa* to other monocot species of the Poaceae family [158], although some specificity is possible. At present, the YSL gene family also includes 18 representatives identified in *Z. mays* and 19 representatives identified in *Brachypodium distachyon* [185]. Among the latter, for example, *BdYSL3* expression was significantly upregulated by Cu deficiency in roots, stems, mature leaves, flag leaves, and reproductive organs, but not in young leaves, where *BdYSL3* transcript levels were highest under normal conditions. Cell-type-specific expression analysis demonstrated that *BdYSL3* was mainly expressed in the phloem of leaves and nodes, which is associated with the involvement of this transporter in the loading of Cu ions into the phloem and metal transport to generative organs [157,159]

The expression of *YSL* genes can be induced by the deficiency of some metals and the excess of some others (Table 2). In *Miscanthus sacchariflorus* seedlings, the expression of the *MsYSL1* gene, encoding a plasma membrane-located transporter for Fe(II)–NA and Zn–NA, was observed in all plant organs, with the highest expression level in stems. In roots, its expression increased under an excess of Mn, Cd, and Pb, as well as under Fe, Zn, and Cu deficiency [149]. *TtYSL1* and *TtYSL2* were slightly or strongly up-regulated in the roots of *T. turgidum* treated with Cd or under combined treatment with Cd and Pb, respectively [108].

In dicotyledonous plants, which lack the ability to synthesize phytosiderophores, YSL transporters are involved in the transport of Fe, Cu, Ni, and Cd complexes with NA (Figure 1) [155,169,171,178,179,181,182,183,186], participating primarily in the maintenance of Fe homeostasis [37,187]. In *A. thaliana*, eight YSL transporters have been discovered to date, with the expression levels of *AtYSL1*, *AtYSL2* and *AtYSL3* being reduced by Fe deficiency and induced by Fe excess, despite the similarity of these transporters to ZmYS1 [175,178,179]. Although AtYSL1, AtYSL2, and AtYSL3 are closely related and have similar gene expression patterns, they differ in their activity in planta [177]. AtYSL1 and AtYSL3 have been identified as the transporters of Fe(II)–NA that mediate Fe(II), Zn, and Cu redistribution from shoots and leaves to seeds in *A. thaliana*, though AtYSL3 also has the capacity to transfer Fe(II)–DMA in yeast [176,177]. The transcription factor WRKY12 negatively regulates the Fe intake in *A. thaliana* seeds by inhibiting the expression of the *AtYSL1* and *AtYSL3* genes [188]. Similarly to *A. thaliana*, eight *YSL* genes have now been identified in *Pyrus bretschneideri*, with *PbrYSL4* having an especially high expression in all tissues, particularly during pollen tube growth [189].

As in monocot plants, different YSL transporters in dicot plants have different metal and ligand specificities (Table 2). Yeast functional complementation indicated that AhYSL3.1 transports Fe(III)–DMA, Fe(II)–NA and Cu–NA, while AhYSL1 specifically transports Fe(III)–DMA in *A. hypogaea*. Among the five *AhYSL* genes, *AhYSL3.1* and *AhYSL3.2* were expressed mainly in young leaves and were upregulated in roots by Cu deficiency, but not by Mn or Zn deficiency, while only *AhYSL1* expression was induced by Fe deficiency in roots [155,182]. The localization of AhYSL1 in the rhizodermis and the ability to selectively transport Fe(III) in a complex with DMA, which is formed and excreted by *Z. mays* plants growing in the vicinity of *A. hypogaea* plants [182], is not only a striking example of mutually beneficial cooperation between cereals and legumes, but also the proof of the possibility that some features of Strategy II for Fe uptake may have evolved in dicotyledonous plants.

In (hyper)accumulators, the expression level of *YSL* genes is higher than that in closely related excluders, and, like in cereals, it is largely organ- and metal-specific. For example, the Cd/Ni/Zn hyperaccumulator *N. caerulescens* has three known YSL transporters whose gene expression levels are higher than those in *A. thaliana*. The *NcYSL3* gene is constitutively highly expressed in all organs of *N. caerulescens*. The *NcYSL5* gene is predominantly expressed in shoots, while *NcYSL7* is mainly expressed in flowers and leaves. The presence of Ni, Zn, or Cd in the growth medium did not alter the level of expression of these genes, and only NcYSL3 was able to transport Fe–NA and Ni–NA complexes [181]. In the accumulator *Brassica juncea*, 27 YSLs with similarity to known *A. thaliana* and *N. caerulescens* YSL genes were obtained. The expression of *BjYSL6.1* and *BjYSL5.8* was found specifically in Cd-treated shoots and Pb-treated roots which might be essential for metal tolerance [190]. The SnYSL3 transporter in the Cd hyperaccumulator *Solanum nigrum* is involved in Fe(II)/Cu/Zn/Cd–NA transport, and the *SnYSL3* expression level was up-regulated by Cd and Fe excess as well as Cu deficiency [183].

In non-graminaceous species, the localization of YSLs is often associated with conducting tissues (Table 2). In situ hybridizations were performed to identify the *NcYSL* expression patterns in *N. caerulescens*. In young roots, the expression of the *NcYSL3* and *NcYSL7* genes was observed in the root central cylinder cells, with *NcYSL7* being predominantly expressed in the pericycle. In older roots, both genes were expressed in the cells surrounding the xylem, and *NcYSL7* expression was also detected in the phloem of both roots and shoots [181]. In *S. nigrum*, in situ RNA hybridization localized the *SnYSL3* transcripts predominantly to the root rhizodermis and stem epidermis, as well as to the conducting tissues of roots, stems, and leaves [183]. This is consistent with the results obtained for *A. thaliana*, in which *AtYSL1* and *AtYSL3* were expressed in the cells of leaf conducting tissues [175,180], while *AtYSL2* expression was detected in the endodermal and pericycle cells opposite to the metaxylem vessels, as well as in the root and shoot conducting tissues, particularly in the xylem parenchyma [178,179]. The data obtained for monocotyledonous and dicotyledonous plants confirm the role of YSL transporters in the loading and unloading of conducting tissues (Figure 1). YSL transporters may also transport Fe–NA complexes from the phloem to the surrounding parenchyma cells [180], as well as metal complexes with NA during leaf aging and seed germination [35,175,176,177,178]. The latter is supported by the fact that *AtYSL1* expression was detected in *A. thaliana* seeds, and that the seeds of the *ysl1* mutant were slower to germinate on a low Fe medium and contained less Fe and NA than those of the wild-type plants did [175]. In the *ysl1ysl3* double mutants, a low content of not only Fe, but also Cu and Zn, was observed compared to the wild type, indicating the involvement of YSL transporters in metal loading in seeds [176].

YSL transporters may be involved in intracellular metal transport, as they can be located not only in the plasma membrane but also in the membranes of some organelles. It is suggested that AtYSL4 and AtYSL6 are located in the inner and/or outer membrane of the chloroplast and may be involved in Fe(II)–NA transport from the chloroplast into the cytosol, since the *ysl4ysl6* double mutants of *A. thaliana* accumulated Fe in chloroplasts, whereas increased expression of *AtYSL4* and *AtYSL6* led to a decrease in its content in plastids [191]. Thus, in the chloroplasts, NA may promote the retention of the mobility of Fe. The role of NA and the abovementioned transporters in Fe translocation during plastid differentiation at the stages of embryogenesis and senescence is widely discussed [37,192], but remains incompletely understood, since these transporters were also found in the tonoplast and endoplasmic reticulum membranes [193,194]. Thus, YSL transporters are widely involved in the regulatory network of micronutrient homeostasis in plants, taking part in metal uptake (AhYSL3.1, ZmYS1, HvYS1, and OsYSL15), long-distance transport and/or metal redistribution (AhYSL3.1, AtYSL1/2/3, BdYSL3, HvYSL2, NcYSL3, OsYSL2/9/13/16/18, SnYSL3, and ZmYSL2) (Table 2). The phylogenetic tree of YSL proteins is shown in [149,158].

### 4.2. Secretion of Nicotianamine

Nicotianamine can be secreted by roots, thereby regulating Zn availability to the plant, and thus facilitate Zn hypertolerance. For example, the roots of Zn-hypertolerant *A. halleri* secreted more NA than did the roots of its non-tolerant congener *A. thaliana*, and NA secretion by the roots of the hypertolerant species increased under Zn excess [62]. In *A. halleri*, NA secretion not only reduced Zn uptake (the Zn–NA complex was barely taken up), but also contributed to the maintenance of Fe homeostasis under excess Zn [195]. The transport of NA from the cytosol to the apoplast is mediated by the efflux transporter of NA (ENA1), which belongs to the MFS superfamily and is located in the plasma membrane (Figure 1). Its physiological role was first studied in *O. sativa*. At a sufficient level of Fe, the *OsENA1* gene was mainly expressed in the rhizodermis of lateral roots. The expression level of *OsENA1* strongly increased under Fe deficiency. In the root tip, *OsENA1* expression was detected solely in the rhizodermis, whereas in the basal part of the root, it was also observed in the cortex and central cylinder tissues [196]. It has been suggested that in Fe-deficient cereals, NA and MA may be synthesized in the intracellular vesicles surrounded by a rough endoplasmic reticulum and located near the plasma membrane [144,196,197,198,199]. The NA and/or MA formed in these vesicles are subsequently transported into the cytosol and exported from the cell via OsENA [196]. In *A. thaliana*, these vesicles were not found [199], suggesting possible differences in the NA transport between *O. sativa* and *A. thaliana*, as they utilize different Fe uptake strategies. Two *ZmENA* genes were identified in *Z. mays*, with *ZmENA1* expression being induced and repressed in the shoots under Fe deficiency and excess, respectively. Subcellular protein localization via the transient expression of green fluorescent protein fusions in mesophyll protoplasts targeted ZmENAs to the plasma membrane, tonoplast, endomembranes, and vesicles [58].

The expansion of *OsNAS1*, *OsNAS2*, and *ZmNAS1;1/1;2* expressions to the rhizodermis under Fe deficiency may be associated with the biosynthesis and secretion of MAs or, possibly, NA [97,103]. MA secretion in *H. vulgare* and *O. sativa* follows a diurnal rhythm [144,197,199,200], which, under Fe deficiency, is accompanied by diurnal fluctuations in the transcription levels of the *NAS* and *NAAT* genes in the roots of *O. sativa* [198]. In *H. vulgare*, the transcription levels of about 50 genes associated with methionine metabolism altered during the day and at night [197].

### 4.3. Transport of Nicotianamine into the Vacuole

Using immunohistochemical methods, NA was detected in the vacuoles and, to a lesser extent, in the cytoplasm of the stele cells in the apical part of *S. lycopersicum* roots [201]. In addition, NA was found in the cytoplasm under normal (10 µM) Fe supply and in the vacuoles of leaf and root cells in the elongation zone in Fe-loaded *P. sativum* and *S. lycopersicum* [202], indicating the possible importance of NA-mediated vacuolar sequestration in the detoxification of excessive metal (Figure 1).

Zinc-induced facilitator 1 (ZIF1) transporter, which belongs to the MFS superfamily, is located in the tonoplast and participates in the NA influx into the vacuole (Figure 1) [34,186,203,204]. *A. thaliana* transgenics overexpressing the *ZIF1* gene were characterized by NA accumulation in the vacuoles of root cells, an increase in the Zn concentration in roots, and a decrease in Zn translocation to the shoots compared to the wild type. An opposite trend was observed regarding the Fe concentration in plant organs. The impaired Fe transport was assumed to be due to a decrease in the NA concentration in the cytosol resulting from its increased influx into the vacuole, whereas the decrease in the Zn concentration in shoots was due to Zn accumulation in the vacuoles of root cells [203]. In contrast, due to the elevated NA concentration in the cytosol of root cells of the *zif1-3 A. thaliana* mutant, the formation of Zn–NA complexes resulted in increased metal mobility and transport into the xylem vessels and further to the aboveground organs, which was accompanied by toxicity symptoms [203]. The expression of *AtZIF1* in *A. thaliana* roots is directly regulated by the PYE transcription factor, and therefore, PYE-mediated repression of *AtZIF1* may be important for maintaining Zn homeostasis under Fe deficiency [119]. In *A. halleri*, a constitutive expression of *ZIF1* was observed. Treatment with Zn considerably induced the expression of this gene in roots, whereas Cd treatment significantly induced its expression in the roots and shoots of the plants from the metallicolous PL22 population of *A. halleri* [21,137] as well as in the roots of *T. turgidum* [108]. The expression of *ZIF1* was also enhanced by Fe deficiency in *A. thaliana* [203]. The localization of ZmTOM2 and ZmENAs to the tonoplast indirectly indicates that ZmTOMs and ZmENAs may also be involved in NA/MA sequestration in the vacuole to detoxify metal excess [58].

The possibility of metal binding to NA inside the vacuole is currently debated. It is believed that NA may bind Ni and Zn, but not Fe, in the vacuole (Figure 1), with Ni–NA complexes being more stable than Zn–NA complexes at the vacuolar pH. However, compared to the stability of these complexes in the cytosol, their stability in the vacuole at the pH values of <6 is significantly lower [65,76]. As the concentration of organic acids in the vacuole is quite high, the contribution of NA to metal binding in the vacuole is probably significantly lower than that to symplastic radial metal transport.

### 4.4. Participation of Nicotianamine in Metal Radial Transport in Roots

Nicotianamine can form stable complexes at a cytosolic pH (7.2–7.5) [76]. This allows it to participate in the symplastic radial transport of metals by enhancing their mobility (Figure 1) [35,45,89,91,111,134,205,206]. The formation of metal–chelator complexes in the cytosol may facilitate metal xylem loading as a result of the limited transport of these complexes into the vacuoles of root cells (Figure 1) [31,207,208]. It is also a fundamental mechanism for metal detoxification in the cytosol, resulting in the alleviation of their toxic effects.

### 4.5. Nicotianamine-Dependent Long-Distance Transport of Metals

NA-deficient plants often accumulate less Fe, Cu, Zn, Mn, and Cd in their shoots [89,127,134,203,205,209,210]. On the contrary, elevated NA concentrations in *O. sativa* due to overexpression of *OsNAS1/2/3* genes led to an increase in the concentrations of Fe, Cu, and Zn in grains [129,130,131,211]. The *nas4x* mutant of *A. thaliana*, which is incapable of NA biosynthesis, showed Zn and Mn accumulation around the root conducting tissues and a decrease in the Zn and Mn concentrations in the xylem sap and shoots compared to those of the wild type. Meanwhile, Fe accumulated in the walls of root cortical cells and its concentration in the xylem sap was not altered [212]. In transgenic *N. tabacum* overexpressing the *HvNAAT* gene from *H. vulgare*, the concentrations of Fe, Cu, and Zn in young leaves and flowers decreased, whereas the overexpression of the *HvNAS* gene, on the contrary, led to an increase in the concentrations of Fe and Zn [205]. Overexpression of *MxNAS2* from *M. xiaojinensis* in transgenic *N. tabacum* plants also led to an increase in the concentrations of Fe, Mn, Cu, and Zn in leaves and flowers [106], while transgenic *A. thaliana* overexpressing the *MxNAS3* gene showed high levels of NA, Fe, Zn, and Mn in leaves [105]. Taken together, the data obtained confirm the involvement of NA in the long-distance transport of Fe, Zn, Cu, and Mn (Figure 1). Suppression of *NAS2* led to a decrease in the Cd concentration in *A. halleri* leaves by no more than 25% [127]. Despite the smaller contribution of NA to the accumulation of Cd compared to Zn, its participation in Cd transport is also noteworthy.

Since NA was detected in the xylem and phloem, this directly indicates its participation in the long-distance transport of metals (Figure 1) [59,70,71,213,214,215,216,217,218,219]. Due to the neutral pH value (7.0–8.0) of the phloem sap, the stability of metal complexes with NA is higher in the phloem than in the xylem (pH 5.0–6.2) [76]. The possibility of NA binding to Cu, Zn, Co, and Ni in the xylem sap, and to Fe, Zn, and Cu in the phloem sap has been demonstrated in different plant species [34,49,65,69,70,71,209,215,217]. In *P. sativum*, Zn/Cu/Ni/Co–NA complexes dominated in the xylem sap [71], whereas in *O. sativa*, NA was the major Zn chelator in the phloem sap [69]. In *R. communis*, Zn/Fe/Cu–NA complexes were found in the phloem sap and Cu–NA complexes were detected in the xylem sap [70]. Regarding Ni hyperaccumulators, Ni–NA complexes were detected in the latex of *Pycnandra acuminata* (previously *Sebertia acuminata*) [220], in the roots, shoots, and xylem sap of *N. caerulescens* [48,49,221], and in *Berkheya coddii* [222]. However, more recent studies suggested a possible artifact in the extraction process [138], meaning that the presence of the Ni–NA complex in *B. coddii* was not confirmed. Indeed, during extraction, cell membranes are destroyed and metals can interact with ligands entering the extract from different cell compartments. Ni–NA complexes were also lacking in the xylem sap of the excluder *T. arvense* [49]. It has been proposed that in cereals, DMA may bind Fe in the xylem and phloem [69,218,223], whereas Cu(II)–DMA complexes were detected in the xylem sap of *O. sativa* [224]. The stability of Fe-DMA complexes is maximal at the acidic pH values (3.5–5.5), whereas Fe–NA complexes predominate at higher pH values [35].

The mechanisms of NA entry into conducting tissues have mostly been studied in relation to the transport of Fe and Zn. It has been suggested that the ENA transporter may be involved in NA loading into the xylem in *O. sativa*, but knockout or overexpression of *OsENA1* did not result in any phenotype related to Fe [196], which brings into question its real role. Fe–NA entry into the phloem has been shown for non-graminaceous species, whereas in graminaceous species, Fe enters the phloem mainly as Fe–DMA. In *A. thaliana*, NA is first transported into secretory vesicles by NA efflux transporter1/2 (NAET1/2), belonging to the nitrate transporter 1/peptide transporter family (NPF). Then NA is secreted through exocytosis into the apoplastic space, where it forms Fe(II)–NA complexes, which are subsequently transported into the phloem by AtYSL1/3 [52,176,177,225]. In *O. sativa,* DMA is synthesized from SAM via NA in MA vesicles. DMA may then be secreted through exocytosis similarly to the secretion of NA [52,144] or it may be exported from the vesicles into the cytoplasm and subsequently transported out of the cell via ENA1 [196]. Then Fe(III)–DMA is transported from the apoplast to the phloem by OsYSL18 [52,152].

In addition to Fe loading, NA is involved in Zn loading into conducting tissues [210]. The efficiency of the NA-dependent pathway of Zn xylem loading was higher in the hyperaccumulator *A. halleri* compared to that in the excluder *A. thaliana* due to the increased NA concentration in the roots of the hyperaccumulator [127,134]. The role of NA in the maintenance of metal homeostasis does not appear to differ significantly between hyperaccumulators and excluders. However, elevated NA levels in the roots of hyperaccumulators are an important determinant of hyperaccumulation capacity, which may be related to its involvement in radial metal transport as well as to the loading and unloading of conducting tissues (Figure 1). Importantly, NA promotes Zn hyperaccumulation in *A. halleri* at the level of Zn translocation, irrespective of plant growth conditions [127].

### 4.6. Nicotianamine-Dependent Metal Transport in Shoots 

Nicotianamine is involved in the remobilization of metals in shoots. All signs of Fe deficiency, including interveinal chlorosis in young leaves, were observed in the *chln* mutant of *S. lycopersicum* characterized by a reduced NA concentration due to the mutation in the *NAS* gene [45,84,107,213,226] as well as in the *nas4x-2* mutant of *A. thaliana* lacking the ability to biosynthesize NA [209]. The concentrations of Fe, Zn, and Mn in mature shoot tissues of the *chln* mutants were higher than those in the wild type, whereas in the shoot apices of the mutant reduced metal concentrations were observed, indicating a role of NA in Fe remobilization in shoots [213]. Fe accumulated in the phloem of the *nas4x-2* mutant of *A. thaliana*, pointing to the involvement of NA in Fe remobilization from the phloem to sink organs [209]. Both mutants were sterile, which indicates the importance of NA in plant reproduction [205,209], particularly in providing nutrients to the male gametophyte [209]. Moreover, transgenic *A. thaliana* overexpressing *MxNAS3* had abnormally shaped flowers and elevated expression levels of flowering-related genes (*AtYSL1*, *AtYSL3*, *AtAFDL*, *AtAP1*, *ATMYB21* and *AtSAP*), which is associated with the need to maintain metal homeostasis, primarily that of Fe and Zn, during the development of generative organs [105].

### 4.7. The Role of Nicotianamine in Nitrogen Fixation in Legumes

Recently, it has been shown that NA is involved in maintaining symbiotic nitrogen fixation in *M. truncatula* nodules, demonstrating the role of this chelator in cross-species interactions and symbiosis. The key enzyme involved in atmospheric nitrogen fixation in legumes is the nitrogenase of the symbiotic root nodule bacteria. Interestingly, MtNAS2, located in the root vasculature and in all nodule tissues in the infection and fixation zones, is required for symbiotic nitrogen fixation in *M. truncatula* nodules as indicated by the loss of nitrogenase activity in the insertional mutant *nas2-1* [117]. Specific expression of *LjNAS2* was found in the nodules of *Lotus japonicas*. The level of *LjNAS2* transcripts in the nodules was the highest on the 24th day after inoculation with *Mesorhizobium loti*, and its expression was localized to the vascular bundles within the nodules [227]. These observations indicate that NA may be involved in the intracellular trafficking of Fe, which is highly important for nitrogenase functioning.

### 4.8. Nicotianamine and Metal Tolerance 

It is crucial to unravel the existing relationships between NA levels and plant metal tolerance, as well as the efficiency of metal transport and detoxification. Elevated NA levels in plant tissues resulted in enhanced metal tolerance and, in most cases, also increased the Ni concentration in the leaves of transgenic *N. tabacum* and *A. thaliana* [228,229,230]. Overexpression of the *MxNAS1* and *MxNAS2* genes [104,106] as well as of the *MxNAS3* [105] gene from *M. xiaojinensis* led to the enhanced tolerance of transgenic *N. tabacum* and *A. thaliana* plants to high and low Fe stresses, respectively. Expression of *SaNAS1* from *S. alfredii* in *A. thaliana* increased NA production and promoted Cd or Zn accumulation in roots and shoots as well as its tolerance to both metals [93]. Conversely, a decrease in NA level resulted in a reduced Ni tolerance but had no effect on Zn tolerance in *A. halleri* and *A. thaliana* [89,210] or Cd tolerance in *A. halleri* [210]. However, the expression of *AhNAS2* or *AhNAS3* in *Schizosaccharomyces pombe* and *Saccharomyces cerevisiae*, respectively, promoted their Zn tolerance [90,91]. *OsNAS3* knockouts were hypersensitive to Fe excess, while NA-overproducing *O. sativa* was hypertolerant to it [99]. Therefore, it is evident that NA is involved in the mechanisms of plant tolerance to metal excess, either through metal complexation as such, or though metal sequestration in the vacuole. 

Importantly, metal-induced changes in the NA content can result not only from direct, but also from indirect effects. For example, increased NA production in Ni-treated plants may be a consequence of Ni-induced Fe deficiency [55]. Metal-induced changes in the mineral profile are manifested differently depending on the species and organ under study, as well as on metal concentration and environmental conditions [10,231,232]. SnYSL3, which transports a broad range of metal–NA complexes, including Cd–NA, was shown to increase Cd tolerance in transgenic *A. thaliana* by decreasing Fe and Mn concentrations in roots and increasing their root-to-shoot translocation [183]. Moreover, increased expression of *MsYSL1* from *Miscanthus sacchariflorus*, which encodes a transporter that is not directly involved in Cd–NA transport, partially induced the expression of three *AtYSL* genes and two *AtNAS* genes under Cd stress in transgenic *A. thaliana*, indicating that MsYSL1 participated in the translocation of Fe, Zn, and Mn from roots to shoots, thus increasing Cd tolerance [149]. Hence, NA plays an important role in the mechanisms of plant tolerance to certain metals, the study of which is one of the promising areas for further research.

## 5. Conclusions, Perspectives and Outlook

### 5.1. Theoretical Importance

The prospects for further studies of NA are of both theoretical and practical importance. Despite the fact that the essential role of NA in metal homeostasis has been reliably established, there are still many aspects that require the close attention of researchers. It is well-known that metals enter the cytoplasm of root cells in the ionic form, as well as in the form of complexes with phytosiderophores or NA via transporters with varying degrees of selectivity [30,54,233]. Consequently, there is competition during metal transport via non-selective transporters, as a result of which metal uptake rates may change. It is especially important to take this into account when studying polymetallic stress, which often occurs under natural growth conditions on metal-enriched soils.

In the cytosol, metals can form complexes with available free ligands, among which, in addition to NA, histidine, phytochelatins, metallothioneins and organic acids play important roles as reviewed in [15,30,234,235,236,237]. Though the formation of metal–ligand complexes obviously takes place in the cytosol, there is still no clear evidence of this. This is due to the facts that (i) no cytoplasm isolation techniques retaining the in vivo metal speciation are available so far, and (ii) metal concentrations in the cytoplasm are generally below the detection limit of techniques that can assess metal speciation. As most extraction techniques are too crude for maintaining metal speciation, the direct detection of Me–chelator complexes is restricted to accessible plant fluids such as the xylem sap, the phloem sap, the embryo sac liquid, or the liquid endosperm [31]. When studying metal speciation in planta, the data on the atomic environment and thereby on potential metal ligands can be obtained, for example, via synchrotron-based X-ray absorption spectroscopy or hydrophilic ion interaction (HILIC) or size exclusion (SEC) liquid chromatography, coupled to electrospray ionization (ESI) mass spectrometry (MS)—HILIC-ICP-MS and SEC-ICP-MS [31,51,71,143]. Modern elemental imaging techniques include, for example, micro-particle-induced X-ray emission (PIXE), synchrotron-based X-ray fluorescence microscopy (micro-XRF), scanning/transmission electron microscopy with energy-dispersive X-ray spectroscopy (SEM/TEM-EDS), laser ablation-inductively coupled plasma–mass spectrometry (LA-ICP-MS), nanoscale secondary ion mass spectroscopy (NanoSIMS), autoradiography, histochemical methods, and confocal microscopy using fluorophores [238]. However, designing techniques and research strategies for the direct detection of metal–ligand complexes in plant tissues and cells is crucial for a deeper understanding of metal homeostasis in plants. Essential transition metals may be present in nanomolar to micromolar concentrations in the cytoplasm, and bound to a wide range of organic ligands [66,239]. For example, the concentration of ‘free Zn’ or ‘labile Zn’, which refers to the Zn^2+^ ions not tightly bound to proteins, is rather low. It has been estimated to range between 50 and 500 pM in the cytosol of eukaryotic cells, which is approximately six orders of magnitude lower than the total Zn content of a eukaryotic cell (~100–500 μM). Therefore, most of the Zn is assumed to be bound to low-molecular-weight ligands as reviewed in [239]. Thus, NA and other ligands serve as metal-buffering compounds to sustain the labile pool of metals [31,32,239]. 

Since several ligands can be simultaneously present in the cytosol in different ratios, competition is possible, both between different ligands for one metal, and between different metals for one ligand. The amount of metal bound to a particular ligand depends not only on the strength and stability of the complexes formed, but also on the amount of different ligands in the cell. For example, NA forms stronger complexes with metals than histidine does [30], so at similar NA and histidine concentrations in the cell, metal ions bind to a greater extent to NA than to histidine. However, the ratio can change significantly at relatively high histidine concentrations. Competition between histidine and NA can occur in the roots of some hyperaccumulators, where the content of both ligands can be high [30], whereas in cereals the NA-to-DMA ratio plays a key role in maintaining metal homeostasis [50,58,131]. The ratio between different types of ligands may also vary for different tissues due to the heterogeneity of the localization of the corresponding transporters as well as the uneven distribution of metals over different tissues, or even over cells of the same tissue [30,142]. However, due to the obvious difficulties in visualizing ligands and their complexes with metals, relevant studies are scarce as yet [143].

Metal binding to NA and histidine in the cytosol, along with the low concentration of phytochelatins in hyperaccumulators [30,237], can limit metal entry into the vacuoles of root cells and thus promote their radial transport towards conducting tissues. On the contrary, in excluders with lower NA and/or histidine concentrations in the roots and a higher concentration of phytochelatins, metals enter the vacuoles of root cortical cells and are less efficiently loaded into the xylem vessels [30,237]. Therefore, it is obvious that NA is one of the key ligands involved in metal hyperaccumulation machinery, although its contribution may differ not only between unrelated species that evolved on metalliferous soils on different continents or islands, but also within the same family or genus.

Nicotianamine plays a prominent role in the long-distance transport of metals. However, we still know very little about how NA enters the xylem vessels and about its role in the loading of metals, e.g., Ni. In general, metal–NA complexes are more stable in the phloem than in the xylem, due to the higher pH (7.0–8.0) in the former [76], which makes further studies of the participation of NA in the transport of metals via conducting tissues and their remobilization in shoots theoretically and practically important.

Both a deficiency and excess of various metals can affect the formation of low-molecular-weight ligands in cells to varying degrees, and this effect may differ for different ligands, which, accordingly, will lead to a change in the buffering capacity of the cytosol [30]. There are few data on the combined effects of different metals on the NA-mediated machinery and, in particular, on the expression of *NAS*, *YSL*, and *ZIF* genes and the content of NA [108,240], as well as on other ligands [30,237,241], which makes this area of research quite promising, especially in relation to the problem of polymetallic stress. The regulation of NA biosynthesis is also poorly studied, though it is crucial to understand the NA-mediated mechanisms for maintaining labile metal pools.

### 5.2. Practical Importance

A significant contribution of NA to the maintenance of metal homeostasis as well as to metal transport and detoxification suggests its direct or indirect participation in various physiological processes, the study of which is a promising direction for further research and has certain potential in the development of approaches used in technologies that have practical applications: phytoremediation, bioenrichment and phytomining. For example, the genetic engineering or selection of hyperaccumulator plants with higher NA levels to enhance Ni and Zn translocation into the shoots could be a promising approach to be used in phytoremediation or phytomining.

Nutrient deficiency is known to be a limiting factor that reduces crop productivity and a widespread cause of human disease and child mortality [131,242,243]. One of the ways to solve this problem is biofortification; that is, the development of approaches to increase the level of bioavailable elements, primarily of Fe and Zn, in agricultural plants, mainly in various Poaceae crops [242,244,245]. Nicotianamine could play an important role in enhancing the nutritional value of these plants, being a candidate for biofortification technology [242,245,246,247,248]. Notably, NA can play a pivotal role in Fe trafficking in the mammalian intestine, as Fe(II)–NA can be absorbed by proton-coupled amino acid transporter 1 (PAT1) in the proximal jejunum [53,249]. Nicotianamine-rich foods are proposed to improve the quality of life of patients with Alzheimer disease via relieving hypertension as well as for the enhancement of the learning and memory functions [250].

Numerous studies using transgenic plants have shown that the increase in the expression of *NAS* genes and in the NA and DMA concentrations is an important factor determining Fe and Zn supply to cereal grains [121,131,132,243,244,245,246,251,252,253]. Conventional, agronomic and transgenic approaches have been used for the biofortification of cereals as reviewed by [242,245,254], and NAS-encoding genes have been recognized as valuable targets for generating metal-enriched crops. For example, overexpression of the *OsNAS* genes increased NA, Fe and Zn concentrations in *O. sativa* grains [129,130,131,211,253,255], and rice *glutelin B1* promoter-driven *OsNAS1* increased Fe concentrations in leaves and polished grains [256]. A couple of the most striking examples are a more than 6-fold increase in Fe concentration in the endosperm of transgenic *O. sativa* lines with the enhanced production of NA and Fe storage protein ferritin [257], as well as a 6- and 3-fold increase in the concentrations of Fe and Zn, respectively, compared to the those in wild-type plants as a result of the expression of the *OsNAS2* gene and the transgene *SferH-1* encoding for ferritin from *Glycine max* [258]. Upon the overexpression of *HvNAS2*, Fe and Zn concentrations increased more than 3- and 2-fold, respectively, in polished grains of transgenic *O. sativa* [255]. A significant increase in the concentrations of these elements in grains was also shown for transgenic *T. aestivum* expressing either *OsNAS2* or *PvFERRiTIN*, or both genes [251]. Field studies of *T. aestivum* lines with a similar *OsNAS2* gene cassette showed a ˃202% increase in NA, up to 30% higher Fe and up to 50% higher Zn in the whole grain, although with considerable variation between the years of study and different field locations [248]. Eight constructs specifically expressing dicot ferritins and *OsNAS2* under different combinations of promoters are already being used commercially [259]. For the purpose of biofortification, it was also proposed to use the transgenic approach combining endosperm-specific expression of the wheat vacuolar iron transporter gene *TaVIT2-D* with the constitutive expression of *OsNAS2*. Transgenic *T. aestivum* plants showed the combinatorial effects of each transgene, namely the redistribution of Fe to the endosperm and higher total Zn and NA levels [243]. Taken together, numerous studies proved that the manipulation of *NAS* gene expression can increase Fe and Zn translocation to the grain in cereals and is a feasible strategy for biofortification.

However, it is important to note that the changes in NA concentration can result in a decreased yield, suggesting that there are trade-offs in NA overaccumulation as reviewed in [245]. Future research efforts need to be focused on the functional characterization of *NAS* genes in order to reveal the physiological mechanisms of these trade-offs and to advance the use of *NAS* genes as a tool for biofortification [245].

Different NA-to-DMA ratios regulated by *NAS* and *NAAT* had different effects on the metal content in grains [131]. Interestingly, transgenic lines of *O. sativa* overexpressing *OsNAS1* and *HvNAATb* showed higher Fe and Zn levels in the endosperm, compared to those in the wild type, whereas Cd accumulation therein was limited [260]. Along with other approaches, better understanding of the physiological roles of each gene involved in the biosynthesis of phytosiderophores and the precise roles of NA/DMA in plants can serve as a theoretical basis to optimize metal biofortification in cereals.

Since metal homeostasis depends on the close coordination of low-molecular-weight ligands and membrane transporters [31,233], the latter play an important role in Fe and Zn biofortification [261]. In particular, the manipulation of the *YSL* genes is an efficient strategy for breeding or engineering cereal varieties with enriched metal nutrition. For example, the overexpression of *ZmYSL2* increased the Zn concentration in the caryopses of *Z. mays* by 31.6% compared to that in the wild type [174]. In addition, when *OsYSL2* expression was driven by the sucrose transporter promoter, the Fe concentration in polished rice was up to 4.4-fold higher compared to that in the wild type [163].

Although most studies have been conducted on cereals in order to develop biofortification approaches, the list of species used is gradually expanding. For instance, transgenic *Solanum tuberosum* expressing *AtNAS1* under the control of the cauliflower mosaic virus 35S promoter contained 2.4 times more Fe in tubers than the wild type did, accompanied by an increased expression of *StYSL1*, *StIRT1*, and *StFRO1* [262].

Transgenic plants characterized by enhanced metal tolerance and the ability to accumulate metals in the aboveground organs are often proposed to be used in phytoremediation technologies in order to clean up the environment [229,230]. However, the practical application of this approach, as in the case of biofortification, is limited by the existing risks associated with the use of transgenic plants [263,264]. Nevertheless, the development of various phytoremediation and biofortification technologies, including those without the use of transgenic plants, is an important area for future research, in which considerable attention could be paid to further studies of NA.

## Figures and Tables

**Figure 1 ijms-24-10822-f001:**
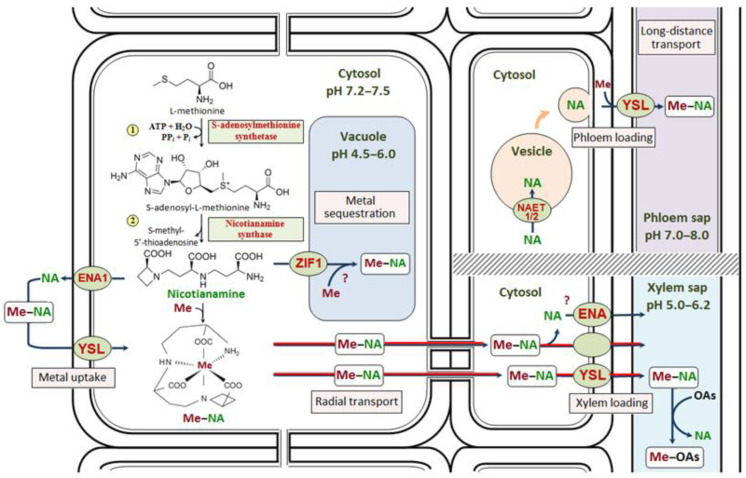
The role of nicotianamine in metal uptake, transport and detoxification in plants. The synthesis of nicotianamine (NA) from methionine is carried out in two steps involving the enzymes S-adenosylmethionine synthetase and nicotianamine synthetase, the latter being present not only in the cytosol but also in various organelles. The NA produced in the rhizodermal cells can be secreted into the rhizosphere via the efflux transporter of NA (ENA1). This is followed by the formation of complexes with metals (Me). In cereals, Me–NA complexes can be taken up by yellow stripe-like (YSL) transporters. Me–NA complexes formed in the cytosol of root rhizodermal and cortical cells can be transported towards the central cylinder, which limits metal entry into the vacuoles of root cells and plays an important role in the mechanism of hyperaccumulation (shown by arrows highlighted in red). Nicotianamine can be transported across the tonoplast via the zinc-induced facilitator 1 (ZIF1) transporter and is probably partially involved in metal binding inside the vacuole, although the stability of NA complexes with metals at the vacuolar sap pH is low. The loading of the Me–NA complexes into the xylem is carried out by YSL transporters. In addition, NA, as a symplastic chelator, can carry out metal delivery to a transporter, e.g., Zn to the P_1B_-type ATPase HMA4. It has been suggested that NA can enter the xylem via the ENA transporter, but this requires further confirmation. In the xylem sap, due to the low pH value, Me–NA complexes are not stable and are partially degraded, and metals bind to organic acids (OAs), forming, for example, citrates and malates. Nicotianamine-dependent metal loading into the phloem in non-graminaceous species occurs with the participation of NA efflux transporters 1/2 (NAET1/2), which provide NA entry into secretory vesicles. Then, NA is secreted through exocytosis into the apoplastic space, where it forms Me–NA complexes, which are subsequently loaded into the phloem via the YSL transporter. In cereals, a similar mechanism involving 2′-deoxymugineic acid, which is synthesized from NA in mugineic acid (MA) vesicles, may function. The stability of Me–NA complexes in the phloem sap is higher than that in the xylem sap due to the higher pH values in the former. As a result, NA participates in long-distance metal transport mainly via the phloem.

**Table 1 ijms-24-10822-t001:** The number of identified nicotianamine synthase (*NAS*) genes in the genome of different species of angiosperms.

Family	Species	Number of *NAS* Genes	References
Brassicaceae	*Arabidopsis thaliana*	4	[87,88,89]
	*Arabidopsis halleri*	4	[90,91]
	*Noccaea caerulescens*	4	[84,87,88]
Crassulaceae	*Sedum alfredii*	5	[92,93]
Euphorbiaceae	*Ricinus communis*	3	[94]
Poaceae	*Aegilops speltoides*	5	[86]
	*Aegilops tauschii*	5	[86]
	*Hordeum vulgare*	9	[95,96]
	*Oryza sativa*	3	[95,97,98,99]
	*Triticum aestivum* (hexaploid varieties)	20–21	[85,86]
	*Triticum monococcum*	8	[100]
	*Triticum urartu*	5	[86]
	*Zea mays*	9–10	[101,102,103]
Rosaceae	*Malus xiaojinensis*	3	[104,105,106]
Solanaceae	*Solanum lycopersicum*	1	[107]

**Table 2 ijms-24-10822-t002:** Most well-studied YS/YSL transporters in monocotyledonous and dicotyledonous plants.

Species	Transporters	Metal Transporting Capacity and Functional Role	Localization of the Transporter Gene Expression or the Transporter Itself	References
Poaceae				
*Brachypodium distachyon*	BdYSL3	Cu in ionic forms; loading into the phloem and transport into generative organs	The expression was significantly upregulated by Cu starvation in roots, stems, mature leaves, flag leaves, and reproductive organs, mainly in the phloem of leaves and nodes.	[157,159]
*Hordeum* *vulgare*	HvYS1	Fe(III) –DMA, Fe(III) –MA; Fe uptake	The expression was enhanced under Fe deficiency in root rhizodermal cells.	[160,161,162]
	HvYSL2	Fe(III)/Zn/Mn/Cu(II)/Co/Ni–DMA, Fe(II) –NA; metal transport	The expression was detected in root endodermis and shoots. The expression in roots was induced by Fe deficiency.	[154]
*Miscanthus sacchariflorus*	MsYSL1	Fe(II) –NA, Zn–NA	The expression was detected throughout the whole seedlings, with the highest level in the stem. The expression was enhanced in roots under Mn, Cd, and Pb excess as well as under Fe, Zn, and Cu deficiency.	[149]
*Oryza sativa*	OsYSL2	Fe(II)–NA, Mn(II)–NA; Fe and Mn transport via the phloem, including translocation into the grains	The expression was induced by Fe deficiency in the phloem cells, especially in the companion cells in leaves, and in mature grains in the epithelium, the vascular bundle of the scutellum, leaf primordium, in the bran and coleoptile.	[116,150,163]
	OsYSL6	Mn(II)–NA; Mn transport	The expression was detected in all root and shoot tissues irrespective of metal deficiency or toxicity.	[153]
	OsYSL9	Fe(III)–DMA, Fe(II)–NA; Fe transport from the endosperm into the embryo in developing kernels	The expression was induced in the vascular cylinder of roots but repressed in the non-juvenile leaves in response to Fe deficiency. At the grain filling stage, the expression was induced in the scutellum of the embryo and in the endosperm cells surrounding the embryo.	[164]
	OsYSL13	Fe transport and redistribution, Fe translocation to the youngest leaves and seeds	The gene was highly expressed in leaves, especially in leaf blades, as well as in both Fe-sufficient and Fe-deficient root cortex. Gene expression was induced by Fe deficiency both in roots and shoots.	[151,156]
	OsYSL15	Fe(III) –DMA, Fe(II) –NA; Fe uptake and transport via the phloem	Gene expression in roots was induced by Fe deficiency, was predominant in the rhizodermis / exodermis and phloem cells under Fe deficiency and was detected only in the phloem under Fe sufficiency. Gene expression was also shown in the flowers, developing seeds, and in the embryonic scutellar epithelial cells during seed germination.	[151,158,165]
	OsYSL16	Fe(III)–DMA, Cu–NA; long-distance transport of metals and metal redistribution, including that of one to the floral organs	Gene expression was detected in the root rhizodermis, root and shoot conducting tissues, leaves, rachilla, palea, lemma, anther and ovary, and was induced by Fe deficiency.	[166,167,168]
	OsYSL18	Fe(III)–DMA; Fe root-to-shoot transport, translocation into generative organs, metal redistribution	Gene expression was detected in generative organs including the pollen tube, and in vegetative organs in lamina joints, the inner cortex of crown roots, in phloem parenchyma and companion cells at the basal part of every leaf sheath. Gene expression in roots and discrimination center was induced by Fe excess.	[152]
*Zea mays*	ZmYS1	Fe(III)/Fe(II)/Cu(II)/ Mn(II)/Ni/Zn/Cd– DMA, Fe(II)/Fe(III)/Ni–NA; metal uptake	Gene expression was localized in root rhizodermal cells and shoots under Fe deficiency.	[35,169,170,171]
	ZmYSL2	Fe–NA, Zn–NA; redistribution of Fe and Zn in kernels	The transporter was localized in the maternal tissue of the basal endosperm transfer cell layer, in aleurone, sub-aleurone, and embryo cells.	[172,173,174]
Brassicaceae				
*Arabidopsis thaliana*	AtYSL1	Fe(II)–NA; Fe long-distance transport and translocation to the seeds	The expression was detected in roots, leaves, flowers, pollen, young siliques, and seeds. The expression in the xylem parenchyma of leaves was upregulated in response to Fe excess. The expression in shoots decreased under Fe deficiency.	[175,176,177]
	AtYSL2	Fe(II)–NA, Cu(II)–NA; Fe and Cu transport	Gene expression was found in the endodermal and pericycle cells opposite to the metaxylem vessels, in root and shoot conductive tissues, and particularly in xylem parenchyma.	[178,179]
	AtYSL3	Fe(II)–NA; Fe long-distance transport	Gene expression was detected in the vasculature of roots and shoots as well as in flowers. The expression in shoots decreased under Fe deficiency.	[176,177,180]
*Noccaea caerulescens*	NcYSL3	Fe(II)–NA, Ni–NA; Fe and Ni transport	The gene was expressed constitutively in all organs, mainly in root central cylinder cells.	[181]
Fabaceae				
*Arachis hypogaea*	AhYSL1	Fe(III)–DMA; absorption of Fe(III)–DMA (DMA is secreted by neighboring cereals)	The expression of this gene in the root rhizodermis was induced by Fe deficiency.	[182]
	AhYSL3.1	Fe(III)–DMA, Fe(II)–NA, Cu–NA; mainly Cu transport	The expression of this gene was detected in the main or lateral roots, but not the root tips, around the conducting tissues, and was upregulated by Cu deficiency.	[155]
Solanaceae				
*Solanum nigrum*	SnYSL3	Fe(II)/Cu/Zn/Cd–NA; metal transport	The expression was detected predominantly in the root rhizodermis and stem epidermis, as well as in conducting tissues of roots, stems and leaves, and was up-regulated by Cd and Fe excess as well as Cu deficiency.	[183]

## Data Availability

Not applicable—no new experimental data were created.

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
