# Peer review of "Nicotianamine: A Key Player in Metal Homeostasis and Hyperaccumulation in Plants"

_ijms, 2023, doi:10.3390/ijms241310822_

Round 1

Reviewer 1 Report

Authors collected the latest data about nicotianamine (NA) such as its biosynthesis, the secretion by plant roots, the mechanisms of intracellular transport of NA and its metal complexes. Moreover, the NA role in metal transport (both radial and long-distance) were discussed. In turn, the biofortification approaches was less described. In my opinion, the article is correctly written. However it will rather dedicated to a small group of specialists than to the wider scientific community.

1.      It is fine that all authors, whose papers were not cited, were apologized. Nevertheless it is better to describe a methodology of papers’ selection i.e. databases and keywords which were used. Please introduce this.

2.      Some papers (especially other latest review articles) could be added e.g. 10.1093/jxb/eraa483; 10.1093/jxb/erab481; 10.3389/fpls.2022.944624

3.      Line 90, wrong date (2042). Please read carefully the text and eliminate similar mistakes.

Author Response

We thank the reviewer for valuable advice and helping us to improve our manuscript. We have enlarged the section dedicated to the biofortification approaches (section 5.2). We believe our review will be of interest for plant physiologists, ecologists, biochemists, agronomists, and biotechnologists studying metal uptake, transport and accumulation in plants as well as metal tolerance mechanisms aimed at maintaining metal homeostasis.

  1. It is fine that all authors, whose papers were not cited, were apologized. Nevertheless it is better to describe a methodology of papers’ selection i.e. databases and keywords which were used. Please introduce this.

We used such databases as Google Scholar, PubMed, Scopus, Researchgate, Web of Knowledge, eLibrary (added to the Acknowledgements section). The key words included, for example, 'low-molecular-weight ligands', 'metal chelators in plants', 'nicotianamine', 'nicotianamine biosynthesis', 'metal ('copper/iron/zinc/manganese/nickel) homeostasis', 'metal tolerance', 'metal transport in plants', 'metal accumulation in plants', 'metal detoxification', 'metal remobilization', 'stress', also in combination, as well as many others. We also performed search by author and checked the reference lists of relevant papers.

  1. Some papers (especially other latest review articles) could be added e.g. 10.1093/jxb/eraa483; 10.1093/jxb/erab481; 10.3389/fpls.2022.944624

 Thank you, we have added these articles (section 5).

  1. Line 90, wrong date (2042). Please read carefully the text and eliminate similar mistakes.

 Thank you, we have corrected this and similar mistakes.

Reviewer 2 Report

The review manuscript entitled with "Nicotianamine: A key Player in Metal Homeostasis and Hyperaccumulation in Plants" has been submitted by Seregin et al. Although the manuscript is comprehensive and relevant in the research area, the main concern is the general structure of the manuscript. It is really difficult to follow to understand for a reader and necessary for major proofreading and editing to scientifically improve its legibility. The manuscript should be reorganized and written to be acceptable for publication. In addition, in terms of the review paper, the authors should provide recent progress as well as the gaps and limitations of research areas related to the theme, thereby conferring the perspectives approaches to solve the remaining barriers in main context.

Overall, the authors scientifically wrote down the manuscript and just require moderate editing of English language.

Author Response

We thank the reviewer for valuable advice and helping us to improve our manuscript. We have reorganized the manuscript by adding a system of sub-headings that make it better structured. In the section 'Conclusion, perspectives and outlook' we address the gaps and limitations of research areas related to the theme and suggest some perspective approaches to solve them. We have enlarged this section. We have also done the proofreading and checked the usage of the English language. What`s more, the manuscript was checked by a colleague fluent in English writing.

Round 2

Reviewer 2 Report

The manuscript has been extensitively improved for the journal publication.

 Minor editing of English language required.